



# Modelling deadwood for rockfall mitigation assessments in windthrow areas

Adrian Ringenbach[1,2,4], Peter Bebi[1,2], Perry Bartelt[1,2], Andreas Rigling[3,4], Marc Christen[1,2], Yves Bühler[1,2], Andreas Stoffel[1,2], and Andrin Caviezel[1,2]

[1]Climate Change, Extremes, and Natural Hazards in Alpine Regions Research Center CERC, 7260 Davos Dorf, Switzerland
[2]WSL Institute for Snow and Avalanche Research SLF, 7260 Davos Dorf, Switzerland
[3]Swiss Federal Institute for Forest, Snow and Landscape Research WSL, Zürcherstrasse 111 8903 Birmensdorf, Switzerland
[4]Department of Environmental Systems Science, Institute of Terrestrial Ecosystems, ETH Zürich, Zurich, Switzerland

**Correspondence:** Adrian Ringenbach (adrian.ringenbach@slf.ch)

**Abstract.** How deadwood mitigates rockfall hazard in mountain forests is a key scientific question to understand the influence of climate induced disturbances on the protective capacity of mountain forests. To address this question both experimental quantification combined with numerical process modelling are needed. Modelling provides detailed insights into the rock-deadwood interaction and therefore can be used to develop effective forest management strategies. Here, we introduce an automatic deadwood generator (ADG) to assess the impact of fresh woody storm debris on the protective capacity of a forest stand against rockfall. The creation of deadwood scenarios allows us to directly quantify their mitigation potential. To demonstrate the functionality of the proposed ADG method, we compare genuine deadwood log patterns, their effective height, and ruggedness at two natural windthrow areas at Lake Klöntal, Switzerland, to their generated counterparts. We perform rockfall simulations for the time a) before, b) directly after and c) 10 years after the storm. The results are compared to scenario d) a complete clearing of the thrown wood, in other words a no forest scenario. We showcase an integration of deadwood in rockfall simulations with realistic, deadwood configurations alongside with a DBH- and rot fungi dependent maximal deadwood breaking energy. Our results confirm the mitigation effect of deadwood significantly reducing the jump heights and velocities for 400 kg rocks. Our modelling results suggest that even after a decade, deadwood has a stronger protective effect against rockfall compared to standing trees. An ADG can contribute to the decision making in forest and deadwood management after disturbances.

## 1   Introduction

Understanding how natural disturbances will affect the protective role of mountain forests against rockfall is an urgent problem in geohazard research. Rock impacts against trees provides a helpful braking effect against falling rocks as successive impacts can significantly reduce rock speed (Dorren and Berger, 2005). Because rockfall transit and deposition zones are often forested, the rockfall-forest interaction has already received considerable attention (e.g., Jahn (1988); Dorren et al. (2005); Kalberer et al. (2007)). Nonetheless it is presently not clear how climate-, natural disturbance and management induced changes in the amount of deadwood will impact the protective effect of mountain forests. While accelerated tree growth and forest expansion near



treeline (Harsch et al., 2009) may contribute to an increased forest protection, this positive result could be offset by increasing rates of natural disturbance and drought related mortality under climate change (Seidl et al., 2017).

Increasing frequency and intensity of natural disturbances is particular likely for forest fires (Mozny et al., 2021; Jain et al., 2020) and bark beetle calamities (Jönsson et al., 2009), but has also been reported for windthrows events in Europe (Seidl et al., 2017) (Fig. 1). Although the influence of climate change on storm intensities and frequencies is presently not clear for the Alps: neither historical data (CH2018, 2018), nor the results among different numerical models or ensembles (Feser et al., 2015; Seidl et al., 2017; Moemken et al., 2018; Catto et al., 2019) do show robust signals on long-term wind-speed

trends for central Europe yet. Nevertheless, a higher windthrow susceptibility also under unaltered storm conditions is likely: Tree anchorage is directly related to soil wetness (Seidl et al., 2017) and due to expected higher winter temperatures and thus fewer days with frozen soil conditions (CH2018, 2018) a lower soil-overturning resistance during the winter storm season is probable. The already reported higher windthrow susceptibility in lower regions (Stritih et al., 2021) may therefore rise with higher temperatures. The growing stock and expanding forest areas due to land-use changes since the mid-$19^{th}$ century are

additionally contributing to an increasing frequency and size of disturbances in the Alps, as windthrow events are more likely in younger, i.e. after 1920, than in older grown forests, originated before 1880 (Stritih et al., 2021). Clearly, rockfall hazard engineers must be prepared in future to understand how large scale disturbances change the protective capacity of mountain forests.

    Researchers have investigated the protection effect of living mountain forests using single tree experiments (Lundström et al.,

2009) and forest (slope) scale experiments (Dorren and Berger, 2005). A primary result of these tests was to find the maximal absorbed rockfall energies (normalized by tree diameter). This allowed rockfall simulation models to be enhanced with single trees (Dorren, 2016; Liu and Li, 2019; Lu et al., 2020; Noël et al., 2021) facilitating the simulation of the protective effect of living forests against rockfall in terms of intensity or risk reduction on single slope (Stoffel et al., 2006; Monnet et al., 2017; Moos et al., 2019, 2020), and on larger regional (Dupire et al., 2016; Lanfranconi et al., 2020) up to alpine-wide (Dupire et al.,

2020; Scheidl et al., 2020) scales. In these approaches the effect of deadwood is largely ignored, although recent experimental studies quantitatively confirm its important role, reporting a reduction of the rock velocity (Bourrier et al., 2012) to stopping rocks completely (Ringenbach et al., 2021).

    In order to address the key scientific questions concerning the role of deadwood in rockfall mitigation, disturbed mountain forests have been monitored for several decades. A significant observational result, based on repeated field surveys of a

windthrow area in Disentis, Switzerland, is that the effective height of the wood steadily decreases over time (Wohlgemuth et al., 2017; Bebi et al., 2015; Marty, 2019). As branches decay, deadwood logs loose their support, resulting in a compaction and loss of height. Declining effective heights have also been reported at the Gandberg site Switzerland. This site was additionally struck by a major bark beetle infestation which caused even more damage than the original storm event (Kupferschmid Albisetti et al., 2003; Caduff et al., 2022). A second important process observed at windthrow areas is the decay of the dead-

wood logs themselves and the associated decrease in load bearing capabilities over time. Studies of the load bearing capacity of deadwood logs have focused on static creep-type loadings from snow pressures (Frey and Thee, 2002; Bebi et al., 2015).





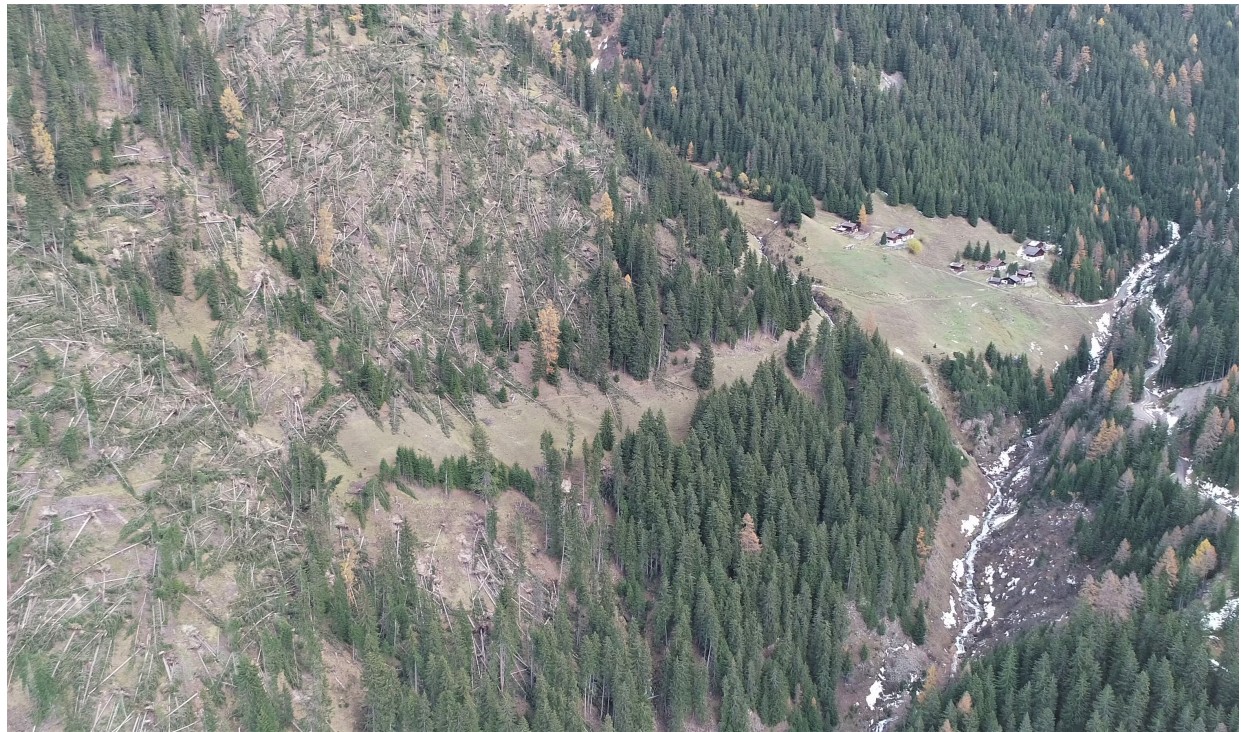

**Figure 1.** The remains of the mountain forest in the $Val\ Tuors$, Switzerland, roughly one week after the windthrow event $Vaia$, 2018. Large areas of the Norway spruce dominated forest suffered windthrow damage and lying deadwood logs are partly piled-up due to the slightly different directions of fall of each stem. Source: UAS-photo SLF, DJI Phantom 4 RTK.

The results of Ammann (2006), however, are better suited for rockfall loadings, as maximal dynamic fracture impact energies of standing snags and their decrease over time were determined.

Discrete element models have been applied to study the impact loading of rocks on wooden structures (Olmedo et al., 2016, 2020). These models are too detailed to be integrated into slope-scale rockfall models. At present the most practical method to incorporate deadwood into rockfall simulations is to adjust the local roughness, slope angle or restitution coefficients to account for fallen tree stems (Fuhr et al., 2015; Costa et al., 2021). This approach may be adequate for de-branched deadwood logs lying on the ground. It misses, however, the often observed fact, that most deadwood logs have no direct ground contact (Kupferschmid Albisetti et al., 2003). This significantly affects the deadwood implementation in rockfall simulation programs in two fundamental ways. The models do not account for the ability for small rocks to roll underneath the deadwood and more importantly, the effective height must be considered to be larger than the deadwood diameter, especially when several deadwood logs are stacked. Both geometrical challenges can successfully be tackled if three-dimensional truncated cones are introduced in rockfall codes, acting as virtual deadwood (Ringenbach et al., 2021). Besides the time-consuming need to manually specify coordinates at each deadwood log end, this method at present lacks a maximal deadwood energy threshold.





These two important, eco-mechanical conditions which vary over time - compression of effective deadwood height, and the reduced absorbing impact energy due to decay - are not sufficiently taken into account in existing deadwood modelling approaches. In order to model the future protective capacity of naturally disturbed forests, these effects must be included in rockfall simulation programs. To accomplish this task, we present an automatic deadwood-generator (ADG), allowing us to generate deadwood logs from a slope- up to a regional-scale. The proposed algorithm allows us to simulate the effects of windthrown trees on rockfall dynamics immediately after a storm event as well as after a given time period, including decay processes and the collapse of the effective deadwood height. Any method to model the distribution and configuration of deadwood in a mountain forest must be calibrated on detailed field observations. The conceptual design of the ADG and its performance is thus examined by applying it to model two sub-areas of an existing windthrow area. The reconstruction capabilities of the ADG with respect to geometrical deadwood configurations, effective heights, and meso-scale surface roughness are compared with the deadwood clusters from field-surveys.

## 2 Methods

### 2.1 Deadwood Generator

The deadwood generator (ADG) integrates deadwood logs into rockfall simulation models. In the following we will specifically couple the python-based, stand-alone software to the RAMMS::ROCKFALL model (Leine et al., 2014; Lu et al., 2019, 2020). The ADG writes the coordinates of the surface points of each overturned stem and root plate into a point cloud file (.pts, see Fig. 2.a). Input variables of the ADG are contained in a so-called treefile, containing $x$- and $y$-coordinates, height and diameter at breast height (DBH) for each standing tree in the upright position. If exact forest surveys are missing, such a file for a specific, representative forest state can be generated in RAMMS::ROCKFALL via the input of forest stand parameters. Additional input parameters for the ADG are: a digital elevation model (DEM), a given main wind azimuth $\alpha$ and its standard deviation $\alpha_{SD}$, the deadwood state (*fresh*, *old*) and the percentage of overturned trees with root plates (blue boxes in Fig. 2.a).

The standard deviation parameter $\alpha_{SD}$ is used to model the variability of the fall direction. When stems fall, they can overlap and pile-up. It is possible that logs lie on top of each other to form barrier-like deadwood stem clusters. Individual directions of fall $\alpha'$ of the logs are a key element to achieve the real observable piled-up logs within the ADG and are normal distributed $\mathcal{N}(\alpha, \alpha_{SD}^2)$. The occurrence of a root plate is objected to a Bernoulli distribution $\mathcal{B}(p_{W_1})$, and $\mathcal{B}(p_{W_2})$ respectively. These factors determine the pivot point, which is located in fall direction from the former stem centre either at the edge of the rootplate, $1.5 \cdot$ DBH (Fig. 2.c), or at the stem base point, $0.65 \cdot$ DBH (Fig. 2.d). Around this point the treetop position is calculated for the different inclination angles $\theta$ according Eqn. 1 - 3:

$$x = x_0 + (H \cdot \sin(\theta) \cdot \cos(\alpha')) \tag{1}$$

$$y = y_0 + (H \cdot \sin(\theta) \cdot \sin(\alpha')) \tag{2}$$

$$z = z_0 + H \cdot \cos(\theta) \tag{3}$$



where $x_0$, $y_0$ and $z_0$ are the coordinates of the pivot point and $H$ is the tree height. The connecting vector between tree top and the stem base - the lower trunk line - is discretized into 100 iso-spaced control points. In case of root plate absence, the pivot points acts as stem base point, otherwise this point is the intercept of the lower trunk line with the root plate in $\alpha'$-direction (defined as in insets c) and d) in Fig. 2). To simulate the tree fall, $\theta$ is inclined in 5° steps. A different minimal compaction

height $H_{\min}$ (green box in Fig. 2) depending of the modeled state ($fresh$ vs. $old$ deadwood) is subtracted from the lower trunk control points in order to accommodate for branches in the felling process and the compacting of the logs over time. At each inclination step, the $z$-values of the shifted control points are compared with their corresponding DEM $z$-values, neglecting those of the lowest 10% and top 20% of the trunk, due to the overturning moment and the increasing flexibility of the trunk top, respectively. $H_{\min}^{fresh}$ and $H_{\min}^{old}$ are set to 0.645 m, resp. 0.258 m, which is a third of the reported effective heights of deadwood

(Bebi et al., 2015), assuming that the trunk, the upward and the downward directed branches contribute equally to the total effective height. As soon as the first $z$-value comparison is negative, that is $z - H_{\min} - z_{DEM} < 0$, the final inclination angle is found and the remaining three-dimensional coordinates of the deadwood log and the root plate are calculated. Further, the resampled DEM (20 cm resolution) is updated with the new $z$-values of the deadwood upper trunk line + $H_{\min}$. This alteration of the DEM results in the generation of an new digital surface model ($DSM_{gen}$) and allows the pilling-up effect since for the

next deadwood log, as it is iteratively updated. The process stops when the entire windthrow tree list is processed. The relative coordinates with respect to the tree top are written in a .pts file, while the absolute coordinates of the tree top is written into a .txt file with the same name.

To directly assess deadwood effectiveness against larger rockfall energies, the fracture impact energy of large-laboratory test data is used: While for fresh deadwood, fracture impact energies are 239 kJ·m$^{-2}$, 8-10 year old deadwood has a reduced

fracture impact energies by 55 % (= 131 kJ·m$^{-2}$) or even 89% (26 kJ·m$^{-2}$), depending on the occurrence of *Fomitopsis pinicola*, a rot fungi (Ammann, 2006). Therefore, its occurrence is assumed according to a Bernoulli distribution, where the probability of occurrence in this study is set to 33 %.

## 2.2 Case study area Klöntal

A comparative study with the windthrow event at Lake Klöntal, Glarus (Switzerland) visualized in Fig. 3, is used to assess

the performance of the ADG. In this case study, a mixed Norway spruce-beech (*Picea abies* (L.) H. Karst - *Fagus sylvatica* L.) forest with a few silver firs (*Abies alba* Mill.) and sycamore maples (*Acer pseudoplatanus* L.) suffered windthrow damage along the south-east facing slopes of Lake Klöntal. During the 26 hours lasting storm event - from 14$^{th}$ to 15$^{th}$ November 2019 - a total of 33 ha were heavily affected. The nearby wind measuring station *Schwander Grat* (IMIS, 2022), located 4.5 km to the southeast, recorded maximal wind velocities up to 25 ms$^{-1}$ (Fig. 2.b). The consolidated 30-minutes-wind directions

were at this ridge position surprisingly stable easterly; however, they feature a mean standard deviation of ±39.8°.

We documented the state of the entire area twice with fixed-wing drone (UAS) surveys: after the windthrow event with the eBee+ RTK (24 January 2020, average flight altitude 200 m above ground) and after the forest service operations with the WingtraOne PPK (25 November 2020, average flight altitude 290 m above ground). Orthophotos with a spatial resolution





**Figure 2.** a) Overview of the automatic deadwood generator (ADG) process, which, based on the input data (blue boxes) and deadwood-state dependent, geometrical premises (green boxes), generates RAMMS::ROCKFALL readable deadwood logs. The contact detection (brown boxes) in case of root plate presence or absence is illustrated in c) or d), respectively. b) Wind directions and maximal wind velocities of the *Schwander Grat* IMIS station, consolidated over 30 minutes periods during the 26 hours lasting storm event (based on Pereira (2022)).





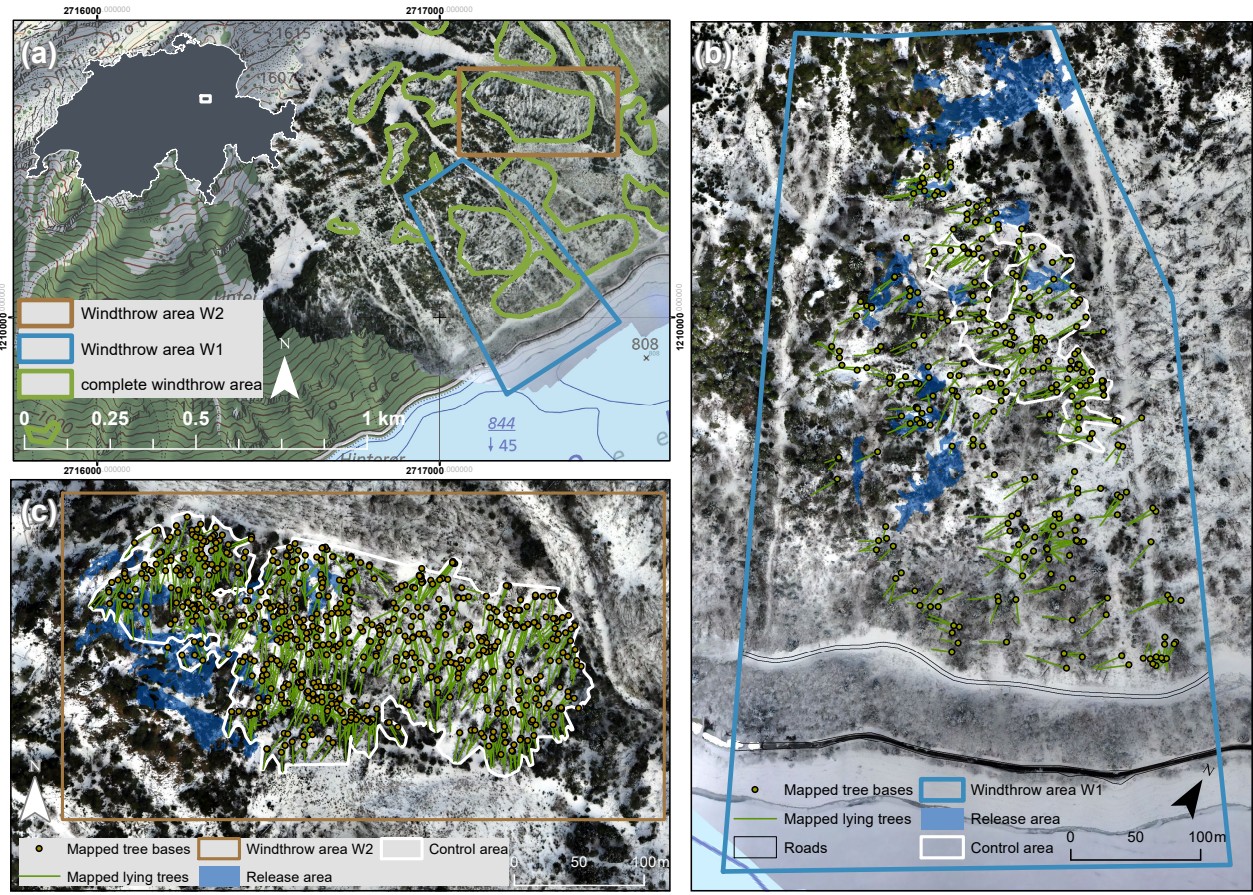

**Figure 3.** a) Overview map of the Klöntal-Case Study, with the orthophoto January 2020 and detail maps of b) windthrow area W1 and c) windthrow area W2 with the mapped lying trees, the control area without standing trees and the used release areas in the rockfall simulations. Source of the topographical map: Federal Office of Topography

of 4 cm and $\text{DSM}_{\text{UAS}}$ with a spatial resolution of 10 cm were produced based on the photogrammetric workflow within the

Agisoft-suite (Bühler et al., 2016).

Two windthrow areas were selected that differ greatly from each other in various aspects: (1) The 8.1 ha large, south-east to south exposed windthrow area W1 is located directly above the road. Here, it is expected that the deadwood reduces the rockfall intensities on the road. Barely any clearing work has been executed during the summer 2020 and thus the influence of the snow load on the deadwood compaction between January 2020 and November 2020 can be assessed. The detailed mapping revealed

a count of 293 logs, yielding a relatively low deadwood density of 36.2 logs $\text{ha}^{-1}$. (2) The second, 6.6 ha large windthrow area W2, with a total of 646 mapped overturned trees and 97.9 logs $\text{ha}^{-1}$ has a nearly three times higher deadwood density as W1. Converted into the more common unit in forestal practice, this results in 58 $\text{m}^3 \cdot \text{ha}^{-1}$ in W1 and 156 $\text{m}^3 \cdot \text{ha}^{-1}$ in W2, based on the volume of the generated logs. W2 is about 400 m to the north-east from W1, but features predominantly east to



south-east aspects and it is located in a side valley, which is not directly above the road. In contrast to W1, almost all of the

dead wood of W2 was cleared in the summer of 2020 to reduce possible bark beetle calamities as the fallen logs implied no direct protective effect for the road.

The overturned trees with DBH ≥ 20 cm were manually mapped based on the UAS-orthophoto within the ArcGIS suite. The starting point of the deadwood line was determined as the trunk base and thus original tree location. The deadwood length was assumed to be the original tree height. Consequently, each log was attributed with their length and diameter, their

type (gymnosperms/conifers vs. angiosperm/deciduous), and falling direction in azimuth. Additionally, the appearance of a root plate was evaluated and if detected, its diameter assessed. The accuracy of all the length measurements account to $\pm 10$ cm. Further quantities of interest are the evaluated vector ruggedness measure (VRM) (Sappington et al., 2007) within a neighborhood of 6 m and the effective heights ($H_{DSM_{UAS}}^{eff}$), being the height difference between the $DSM_{UAS}$ and the official, pre-event, swissALTI3D $DEM_{sA3D}$ (original resolution 50 cm, swisstopo (2018)), both resampled to 20 cm. Those remote

sensing data were compared with the $DSM_{gen}$, whereof the effective height and ruggedness measure, $H_{gen}^{eff}$ and $VRM_{gen}$ were calculated, respectively. While a comparison of the azimuth direction of detected trees and the generated deadwood over the whole area is valid without limitations, $H_{eff}$ and VRM of the $DSM_{UAS}$ are affected by remaining standing trees. Therefore, the control areas with barely no standing trees were defined: 1.1 ha for W1 and 4.5 ha for W2 (see white lines in Fig. 3.b) and c). In order to focus on the modified rather than matching areas, zero values were omitted from the evaluated density function.

In other words, areas where the ADG generator left the input $DEM_{sA3D}$ unaltered were ignored, as the identical DEM was subtracted from the $DSM_{gen}$ in order to get $H_{gen}^{eff}$.

The manual deadwood mapping provides all the necessary input data for the ADG such as the treefile, the main wind direction $\alpha$, its standard deviation $\alpha_{SD}$ and the root plate existence ratio. Hence, the ADG is executed to replicate the mapped scenario (Scenario 0°). Additional deadwood alignments were generated with $\alpha$ -90°, -45°, +45° and +90° while keeping $\alpha_{SD}$

constant.

## 2.3  RAMMS::ROCKFALL simulations

Rockfall simulations were performed with RAMMS::ROCKFALL, a three dimensional rockfall code based on a rigid-body approach (Leine et al., 2014, 2021), which considers compactable soils (Lu et al., 2019), standing trees (Lu et al., 2020) and three dimensional cones as deadwood (Ringenbach et al., 2021). We used the official swissALTI3D $DEM_{sA3D}$, in its initial res-

olution of 0.5 m (swisstopo, 2018). The used soil parameters for the majority soil class $forest$ were $M_E = 1.8$ and $C_d = 1.4$. Roads, as depicted in Fig. 3.b), $M_E = 75$, $C_d = 2.0$ and the lake, $M_E = 0.1$, $C_d = 10000$), were assigned its own soil parameters.

To obtain realistic starting positions within the automated potential release area approach, we deployed a simplified slope angle distribution (SAD) analysis of the region (see blue marked areas in Fig. 3.b and c). We abstained from decomposing

the SAD into several Gaussian distributions representing different, generalized slope classes (Loye et al., 2009) due to inconsistent results. The threshold angle was simply defined as the inflection point of the regional SAD (= 43°) as lower bound for starting point areas. For a jitter suppression and reduction of starting points, a weighted focal analysis within a 5x5 cell





focal-neighbourhood for cells with a slope above 43° is performed. Only release areas, which contribute to rockfall towards the deadwood area, are taken into account. In W1 378 and in W2 278 starting points were finally used with an approximate

distance between the starting points of 7.25 m (regular grid assumed). One mass class of 400 kg or 0.15 m³ and two shape classes platy and compact - according to the Sneed and Folk (1958)-classification - made up by the perfectly symmetrical EOTA rock shapes (ETAG 027, 2013) were used for the simulations. Each rock was released with 10 different orientations, which leads to a total of 7560 trajectories in W1 and 5560 trajectories in W2 for each azimuth scenario. Both deadwood states - $fresh$ and $old$ - for the five $\alpha$-azimuth scenarios - shortened as e.g. $DW_{fresh}^{\alpha}$ - are computed. Additionally, a comparison

with the undisturbed, $original$ forest and the scenario with the $cleared$ area without trees were calculated for both areas, resulting in 24 RAMMS::ROCKFALL scenarios. Figure 4 exemplary visualize RAMMS::ROCKFALL simulations for W2 for the four scenarios $original$ forest, $cleared$, $DW_{fresh}^{0°}$ and $DW_{old}^{0°}$. To evaluate the vast data set, the 95-percentile raster values of the kinetic energy $E_{KIN}$, jump height $j_H$ and velocity $v$ within the coloured frames in Fig. 3.b) and c) are compared by evaluating each $DEM_{sA3D}$ raster cell for the given parameter. Overall changes in kinematic behaviour are then deduced from

this two-dimensional screening. Additionally, the deposition altitudes of the rocks are analysed and the total reach probability ($P_r$) for the evaluation lines (yellow lines in Fig. 7) calculated.

## 3 Results and discussion

### 3.1 Mapped deadwood logs

Of the 939 manually mapped lying deadwood logs, a total of 512 trees had a visible overturned root plate: 44% in W1 and 55%

in W2, yielding $p_{W1} = 0.44$ and $p_{W2} = 0.55$ for the used Bernoulli distributions. Thereof 411 were coniferous (W1: 111, W2: 300) and 101 deciduous (22/79). A linear regression between the DBH and the root plate diameter in W1 and W2 and for the two tree types are in the range between 2.7 - 4.4 · DBH, while the deciduous feature on both areas steeper gradient 3.6 - 4.4 · DBH, as the coniferous, 2.7 - 3.8 · DBH (Fig. 5). These allometric relationships are similar to the reported regression equitation of root-pits in the review by Robbins et al. (2014): $D = 0.039 \cdot DBH + 0.439$ with $R^2 = 0.64$, based on worldwide, in total 676

in field measurements, of which 70 measurements were conducted in Austria, with the same tree species as in this study, beside the abundance of sycamore maples. The linear regression, however, has limited descriptive power ($0.114 \leq R^2 \leq 0.283$), which might originate due to several large root plates formed by multiple trees, and the difficulty to attribute those root plate diameters to the single trees. Additional inaccuracies occur from the complexity to perform the nearly vertical root plate measurements in the orthophoto. Fusing the values from literature with the own measurements, the assumed $RP_{\varnothing} = 3 \cdot DBH$ is certainly

not overestimating the diameter of root plates despite the large uncertainties. A different, yet to be verified $RP_{\varnothing}$ distribution function might be more suitable to depict the DBH-$RP_{\varnothing}$ correlation in future studies, judging by the wide variation in both, the own data and the values from literature.

Despite the proximity of the two windthrow areas to each other, the measured deadwood fall directions differed considerably between W1 and W2 with main wind directions $\alpha_{W1} = 26.0° \pm 48.2°$ and $\alpha_{W2} = 7.2° \pm 27.0°$. The topography clearly disturbed

the wind systems and affected the main deadwood falling direction. In order to focus the central part of the distribution (Fig.

Earth **Surface**
Dynamics
Discussions



**Figure 4.** Simulated velocities of the four main scenarios within windthrow area W2: a) *original forest*, b) the *cleared* area, c) the deadwood-scenario with the highest mitigation effect ($DW_{fresh}^{0^{\circ}}$) and d) the corresponding $DW_{old}^{0^{\circ}}$ scenario



Earth **Surface**
**Dynamics**
Discussions

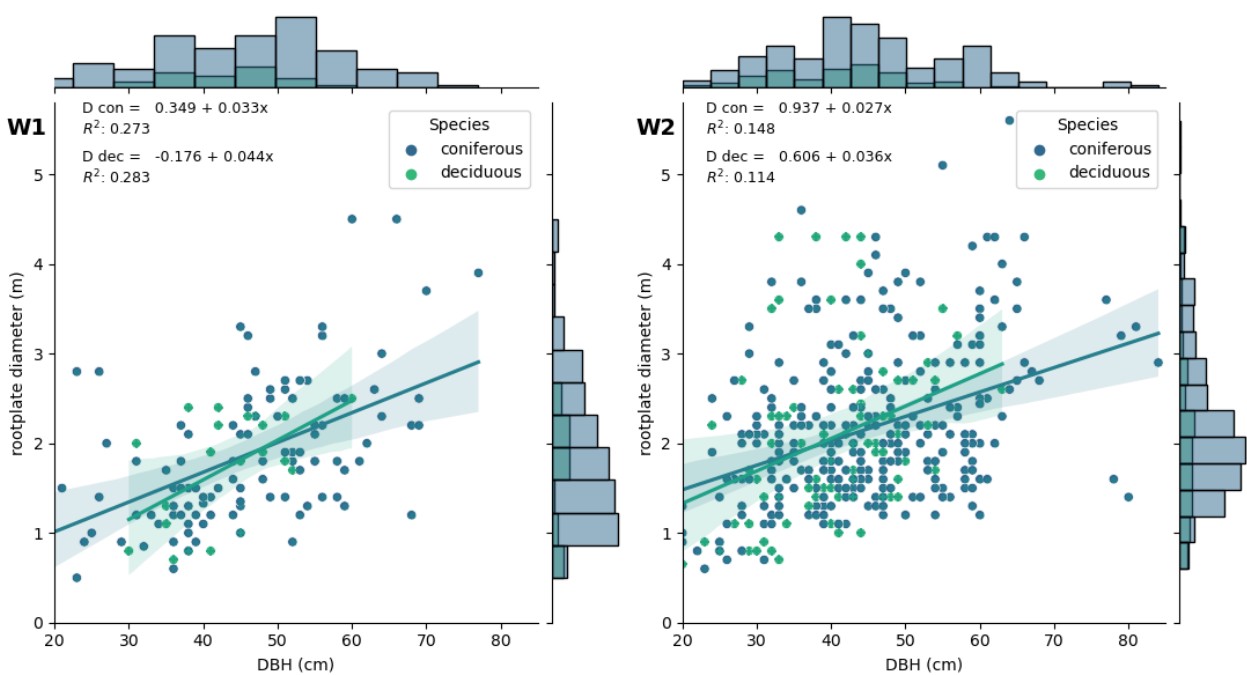

**Figure 5.** DBH - root plate diameter ratio of the overall 512 mapped lying logs with a visible overturned root plate. The ratio between coniferous and deciduous in area W1 (left) is 111/22, while it is 300/79 in W2 (right). Although the linear regressions between the DBH and the root plate diameter has a poor descriptive value ($0.114 \leq R^2 \leq 0.283$), all the calculated slopes of the linear regressions of the two areas and two tree types are within the range between 2.7 - 4.4 · DBH, while the deciduous show on both sites steeper slopes

6.a, d), the values for the standard deviation for the generated deadwood logs needed to be slightly reduced, $\alpha_{W1} = 26°\pm 38°$ and $\alpha_{W2} = 7°\pm 24°$, arguing that the contributions at the extreme end of the deviation scale mainly occur due to chaotic boundary effects during the felling, pile-up process which are not built into the ADG. In the future, the mapped deadwood data set might serve as machine learning training data, aiming for an automatic deadwood log identification.

## 3.2 Observed and generated effective heights $H_{\text{eff}}$

In both areas, the observed $H_{\text{eff}}$ distributions deducted from the second drone flight $DSM_{\text{UAS}}^{\text{Nov}}$ show that over 75% of all $H_{\text{eff}}$ values are rather low (<1 m, Fig. 6.b, e and Tab. 1). For better readability $H_{\text{eff}}$ based on a given DSM will be labelled as follow:

$$H_{\text{eff}}\left(DSM_{\text{UAS}}^{\text{Nov}}\right) \equiv H_{\text{UAS}}^{\text{Nov}},\ H_{\text{eff}}\left(DSM_{\text{gen}}^{\text{fresh}}\right) \equiv H_{\text{gen}}^{\text{fresh}},\ \dots$$

However, the development over time differed between the areas, as $H_{\text{UAS}}^{\text{Jan}}$ shows: While in W1 the ratio of low $H_{\text{eff}}$ values raised from 69.7% in January by just 5.4% to 75.1%, this share raised in W2 from 29.1% by 50.4% to 79.5%, accompanied by the abrupt decrease of the amount of tall deadwood, $H_{\text{eff}} \geq 2$ m, from originally 35.7% to 8.0%. In both areas, the $H_{\text{gen}}^{\text{fresh}}$ is more right-skewed with generally higher $H_{\text{eff}}$ values than the $H_{\text{gen}}^{\text{old}}$. In W2, the $H_{\text{gen}}^{\text{fresh}}$ aligns well with the observed $H_{\text{UAS}}^{\text{Jan}}$,





especially for values $\geq 2$ m, as both distributions comprise at least 35.7% of such tall deadwood. Such an agreement could not be found in W1, where both $H_{gen}^{fresh}$ and $H_{gen}^{old}$ overestimate tall deadwood compared to both drone based $H_{UAS}$: While tall
deadwood in $H_{gen}^{fresh}$ has a share of 40.0% and comprises still 17.7% in $H_{gen}^{old}$, solely 8.8 % of all $H_{UAS}^{Jan}$ values are $\geq 2$ m.

We consider the main discrepancy of the observed $H_{UAS}^{Jan}$ values $\geq 2$m between the areas mainly due to the deadwood density. Higher densities as in W2 are accompanied with more log stacking, which leads to higher effective heights. If the deadwood stems are too far apart, they will not pile-up on each other and lie on the ground instead, which is associated with lower $H_{eff}$ values. The observed unification by the second flight, on contrast, is due to a slight deadwood compaction in W1 and
the extensive clearing in W2. The reduction between $H_{gen}^{fresh}$ and $H_{gen}^{old}$ affirms the expected influence of the minimal deadwood height $H_{min}$ on the resulting $H_{eff}$, what makes it a decisive variable. The windthrow event, from where the default $H_{min}$ used in this study were adapted from, exhibited with 363 logs ha$^{-1}$ (Bebi et al., 2015) an even higher deadwood density as W2, which explains the observed $H_{eff}$ overestimation therein. If forest with lower windthrow densities are modelled in the future, the $H_{min}$ variable should be adapted and reduced.

## 3.3 Roughness-evaluation based on VRM comparison

Similarly to the naming of the effective heights, the vector ruggedness measures are abbreviated based on its origin from the different DSM:

$$\text{VRM}\left(\text{DSM}_{UAS}^{Jan}\right) \equiv \text{V}_{UAS}^{Jan}, \ \text{VRM}\left(\text{DSM}_{gen}^{fresh}\right) \equiv \text{V}_{gen}^{fresh}, \ \cdots$$

Within W1, the density distribution of $V_{UAS}^{Jan}$ and $V_{UAS}^{Nov}$ remains nearly unchanged (Fig. 6.c and Tab. 1). Solely a VRM-peak
shift from 0.07 to 0.06 is visible. In contrast, in W2 the two $V_{UAS}$ differ distinctly (Fig. 6.f): the initial bipolar distribution pattern of $V_{UAS}^{Jan}$ changed into the left-skewed pattern of $V_{UAS}^{Nov}$. The effect can be explained due to the removal of the deadwood in the area W2 between the two DSM acquisition dates and is in line with findings of Brožová et al. (2021) that deadwood increases the VRM. The deviations to their slightly higher deadwood VRM values (VRM = 0.17 - 0.30) can be explained by a different DSM resolution (1.0 m) and a larger moving window size (49 m$^2$) and matches with the presented sensitivity
analysis. The influence of the stand parameters of the windthrown Norway spruce-larch forest cannot be assessed, since no information on the stem density was given. In W1, both ADG-based DSM are smoother than their corresponding UAS-based DSM, as $V_{gen}^{fresh}$ and $V_{gen}^{old}$ underestimate high VRM Values in comparison to their UAS counterparts (Tab. 1). In contrast, the generated VRM values in W2 result closer to the observed ones. In W1, the ADG-based $V_{gen}^{fresh}$ underestimate in general the VRM, compared to the UAS-derived $V_{UAS}^{fresh}$, since low VRM values are over-represented (78.6%, vs. 50.2%) and high VRM
values are under-represented (3.0%, vs. 12.3%, see Tab. 1). Such ratios, even more apart, are also reported for $V_{gen}^{old}$ and $V_{UAS}^{Nov}$.

Three presumed issues play a central role: (1) The simplified deadwood representation in $\text{DSM}_{gen}$, which neither considers branches nor large stem diameters, is sufficient to represent the piling-up effect but has its shortcomings when assessing the VRM in detail. (2) The harmonization of different input quality can have a negative effect: despite the resampling to 0.2 m, there might be a difference in the level of detail between the coarsened $\text{DSM}_{UAS}^{Jan/Nov}$ and the upsampled $\text{DEM}_{sA3D}$ - used as





DSM$_{\text{gen}}$ basis - within the moving window size of 6 m$^2$. (3) The lower the deadwood density, the more pronounced the effects of (1) and (2) come into play. We attribute the better match in W2 caused by the higher deadwood density.

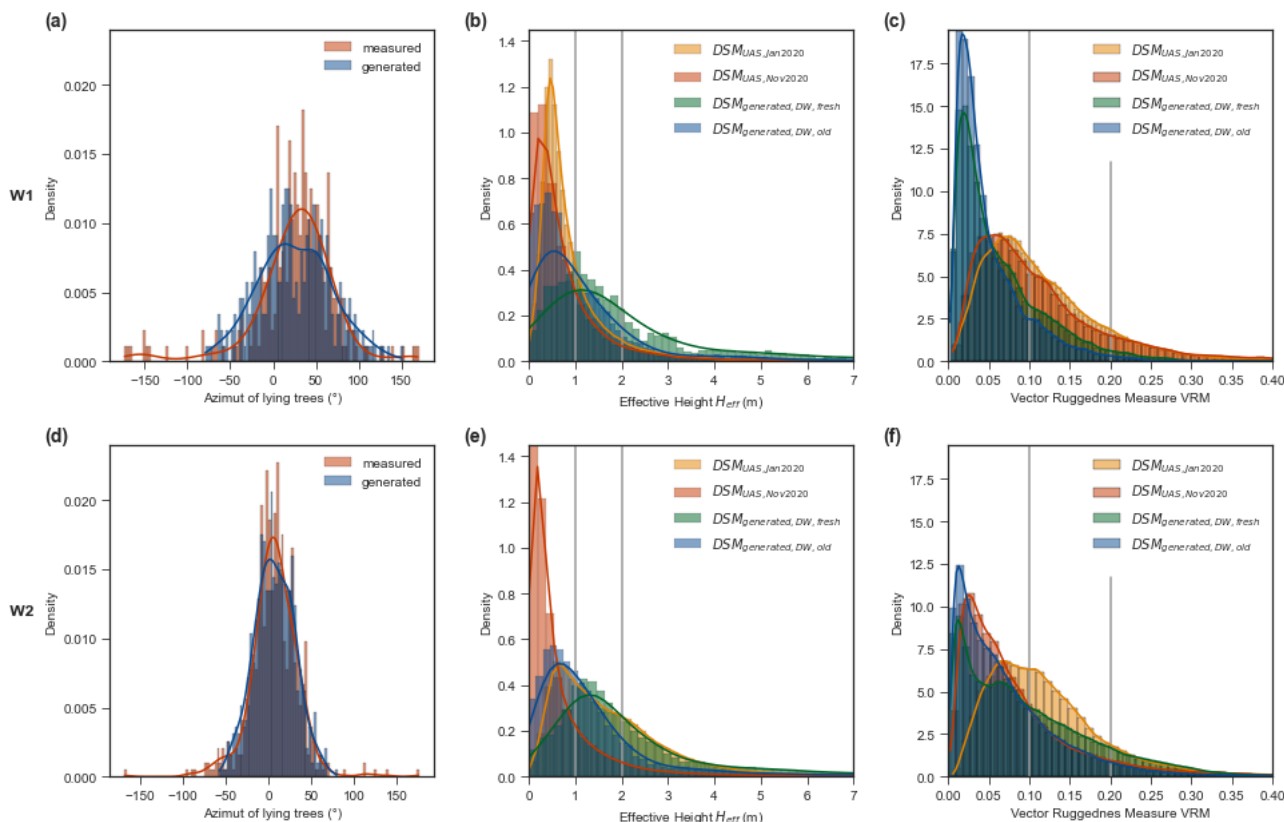

**Figure 6.** Three descriptive comparison indexes between the UAS-observed and ADG generated deadwood-composition for the control area (according Fig. 3) of W1 (top) and W2 (bottom): Azimuth of the falling tree (a, d), effective height H$_{eff}$ (b, e) and vector ruggednes measure VRM (c, f). Vertical grey lines indicate the location of the comparison in Tab. 1.

### 3.4 Deadwood influence on simulated rockfall dynamics

For both areas resulted a significant reduction of the rockfall affected area due to deadwood (Fig. 7). To quantify the rockfall stopping capacity of deadwood the total reach probabilities P$_r$ at given evaluation levels (yellow lines in Fig. 7) are assessed

in Table 2. The relative P$_r$ reduction $\Delta_{P_r}$ of the disturbed and undisturbed forest with respect to the *cleared*, no forest, scenario serves as generalized reduction performance index. The two deadwood states *fresh* and *old* obstruct more rocks as the *original*, pre-windthrow forest and the *cleared* scenarios, whereby this exact ranking order is valid for both areas. While in both areas $\Delta_{P_r}$ of the *original* forest is similar, the mitigation effect of the deadwood is higher in W2, especially for *fresh*, but also for *old* deadwood. The highest mitigation effects are observed for $DW_{fresh}^{45^\circ}$ in W1 with only 13.7% of the rocks reaching

the evaluation level, 11.5% for $DW_{fresh}^{0^\circ}$ in W2, respectively. While under *cleared* conditions, with P$_r$(W1) = 55.0%, P$_r$(W2)



**Table 1.** Percentages of low ($H_{eff}$ < 1 m; VRM < 0.1) and high ($H_{eff}$ > 2 m; VRM > 0.2) values of the two key comparison parameters, derived from the different digital elevation models per windthrow area W1 and W2.

|  |  |  | $DSM_{UAS}^{Jan}$ |  | $DSM_{gen}^{fresh}$ |  | $DSM_{UAS}^{Nov}$ |  | $DSM_{gen}^{old}$ |  |
|---|---|---|---|---|---|---|---|---|---|---|
|  |  |  | W1 | W2 | W1 | W2 | W1 | W2 | W1 | W2 |
| Heff | < 1 m | (%) | 69.7 | 29.1 | 26.4 | 18.6 | 75.1 | 79.4 | 56.1 | 44.9 |
|  | > 2 m | (%) | 8.8 | 35.7 | 40.0 | 46.7 | 9.2 | 8.0 | 17.7 | 23.7 |
| VRM | < 0.1 | (%) | 50.2 | 45.3 | 78.6 | 59.3 | 54.8 | 70.7 | 87.0 | 74.1 |
|  | > 0.2 | (%) | 12.3 | 12.4 | 3.0 | 12.8 | 13.1 | 8.4 | 1.4 | 5.3 |

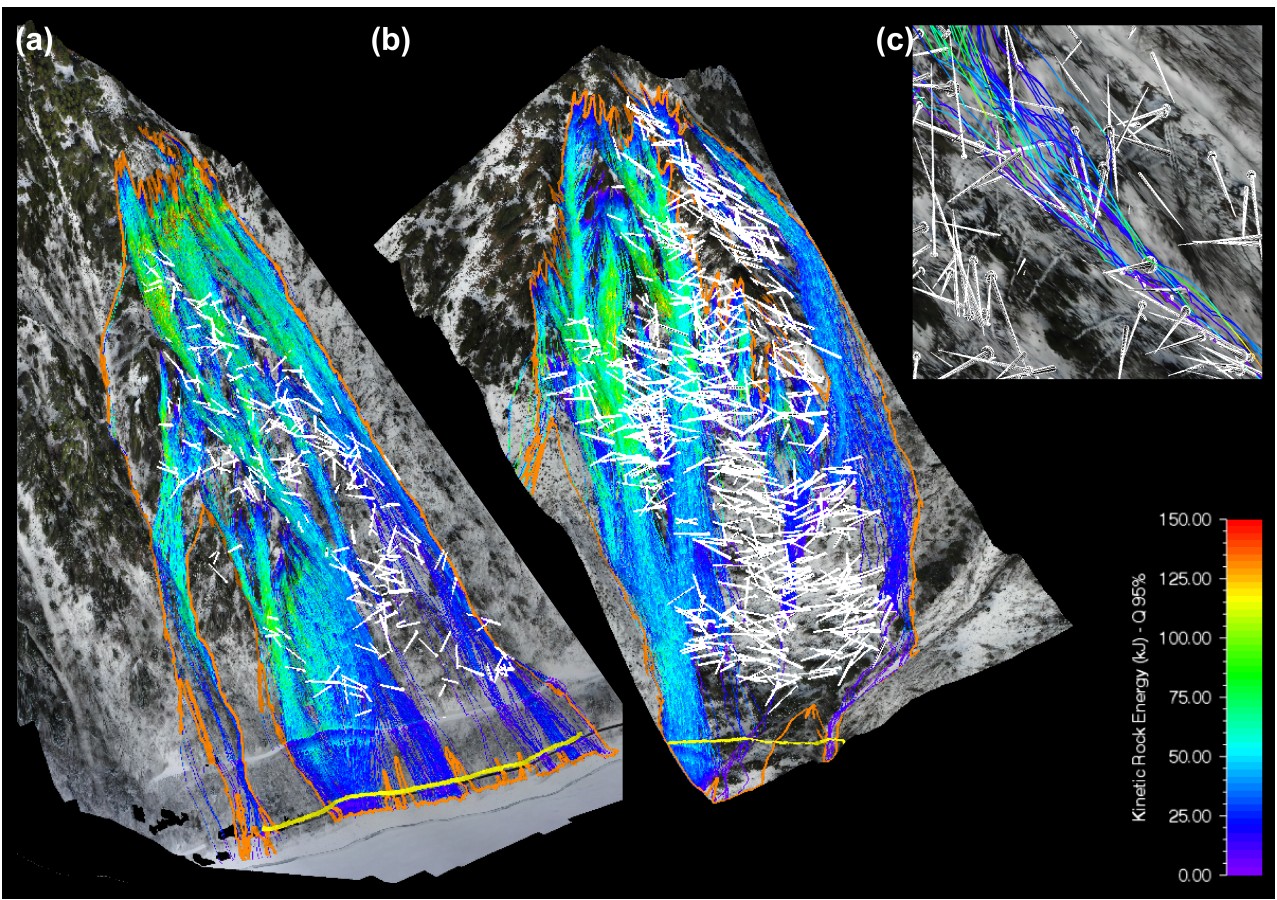

**Figure 7.** Simulated kinetic rockfall energy of the deadwood-scenario with the highest mitigation effect (rainbow color covered area) compared to the the outline of the *cleared*-scenario (orange plotted line): a) Wind throw area W1 with $DW_{fresh}^{+45°}$ and b) Wind throw area W2 with $DW_{fresh}^{0°}$, both with the reach probability evaluation line, plotted in yellow. c) a close-up of a selected subset of simulated trajectories visualized three-dimensionally interacting with the uprooted, piled-up deadwood logs and root plates.





= 90.7%, up to nine out of ten released rocks reach the base of the slope, $\Delta_{P_r}$ for the best performing DW scenarios, indicate that in W1 75.1% of those rocks are retained, and even 87.3% in W2. The elevations of the deposited rocks along the slope are evaluated to gather further insights of the deposition patterns (Fig. 8. a, b). While W1 features a rather uniform deposition pattern along the slope, in W2 the rocks are particularly deposited at elevations between 1300-1260 meters and 1240-1140 m

a.s.l. for the better performing DW scenario. The absence of those features in certain DW alignments is additional proof of the relevance of the direction of fall in the ADG.

**Table 2.** Total reach probabilities $P_r$ along the evaluation lines (according Fig. 7) for the scenarios *original* (*ORG*), *cleared* (*CLR*), *fresh* and *old* deadwood and the relative $P_r$ reduction (=$\Delta_{P_r}$), compared to the *cleared* scenario.

| | | | ORG | CLR | $DW_{fresh}$ | | | | | | | $DW_{old}$ | | | | | | |
|---|---|---|---|---|---|---|---|---|---|---|---|---|---|---|---|---|---|---|
| | | | | | -90 | -45 | 0 | 45 | 90 | mean | SD | -90 | -45 | 0 | 45 | 90 | mean | SD |
| $P_r$ | W1 | % | 42.5 | 55.0 | 35.1 | 27.4 | 21.9 | **13.7** | 30.1 | 25.7 | 7.3 | 38.3 | 49.4 | 37.4 | **32.9** | 38.1 | 39.2 | 5.5 |
| | W2 | % | 70.1 | 90.7 | 41.2 | 28.2 | **11.5** | 13.8 | 35.2 | 26.0 | 11.7 | 57.1 | 47.7 | **26.2** | 41.5 | 56.2 | 45.7 | 11.3 |
| $\Delta_{P_r}$ | W1 | % | 22.7 | 0.0 | 36.2 | 50.2 | 60.2 | **75.1** | 45.3 | 53.3 | 13.3 | 30.4 | 10.2 | 32.0 | **40.2** | 30.7 | 28.7 | 10.0 |
| | W2 | % | 22.7 | 0.0 | 54.6 | 68.9 | **87.3** | 84.8 | 61.2 | 71.3 | 12.9 | 37.0 | 47.4 | **71.1** | 54.2 | 38.0 | 49.6 | 12.5 |

The kinetic energy, translational velocity and jump height plots (Fig. 8. c-h) are all characterised by the same general pattern: The intermediate value ranges of each parameter of interest is diminished, generally in the scenario order *cleared*, *original*, *old* to *fresh* deadwood - with some subtleties -, while the extremal parts of the parameter ranges are hardly affected by scenario

choice. Differing effects between *original* and *old/fresh* deadwood scenarios are governed mainly by two opposing trends: lower impact probability of standing, living trees with higher energy threshold vs. higher impact probability but with a lower energy threshold in the presence of lying deadwood. This is well illustrated in the inset of Fig. 8.d. While at low energies the *original* forest scenario only the *cleared* scenario is inferior in terms of mitigation effect, the relative effectiveness compared to other scenarios increases at higher total kinetic energies $E_{KIN}$, until the *original* forest finally becomes the superior state at

$E_{KIN}$ > 85 - 90 kJ. This transition fits well with the expected maximal energy absorption of a fresh spruce deadwood log with a diameter of 67 cm of 84 kJ (Ammann, 2006), which corresponds to the $99^{th}$ DBH-percentile in the observed mountain forest. In contrast, an equivalent, living tree has an expected $E_{KIN}$ absorption up to 618 kJ (Dorren and Berger, 2005). In order to assess the aged deadwood configuration, the same log is assigned a maximal kinetic energy absorption of 46 kJ resp. 9 kJ, depending on a *fomitopsis pinicola* occurence (Ammann, 2006). The comparison between *old* and *cleared* scenarios highlights, that

mitigation potential does not solely rely on the maximal possible deadwood energy absorption: $DW_{old}$ configurations generally feature higher mitigation effects up to ~ 55 kJ than the *original* forest. The reduction in number of such high energy grid cells is likely due to stopping processes further up-slope at lower velocities, preventing many trajectories reaching high energy regimes.



**Figure 8.** Resulting rockfall simulation of area W1 (left column a, c, e, g) and W2 (right column b, d, f, h) summarized in several kernel density estimate plots: A comparison according to the elevation a.s.l. of the rocks deposition points (a, b), the distribution of kinetic energy (c, d), velocity (e, f) and jump heights within the colored frames according Fig. 3.b and c). The zero point density overshoot in c) - h) arise from unimpaired raster cells within the evaluation areas, i.e. no rockfall trajectory is passing through. An exclusion would be inaccurate, as otherwise the differences between the scenarios would be distorted, since different numbers of raster cells are involved.





### 3.5 Linking rockfall mitigation to forest management

Our study confirms that deadwood plays a significant role in reducing the risk of rockfall during the first years after a natural disturbance. It equally highlights the complexity of underlying processes and management of deadwood after such events. How the protection effect of deadwood is developing over longer time periods and is complemented by early successional stages of post-disturbance forests, has to be investigated by additional studies, including experiments and models in more advanced stages of wood decay after disturbances and systematic observations of secondary rockfall releases behind decaying wood. Long-term observations and modelling of snow avalanches after natural disturbances suggest a minimum of the protection effect after 10-15 years, followed by an increase due to the combined effect of post-disturbance regeneration and the remaining terrain roughness of dead wood (Caduff et al., 2022). However, this effect may strongly vary according to site conditions, the initial stand structure and species composition of a forest before the disturbance and external management factors such as partial removal of deadwood, planting of additional trees or browsing (Bebi et al., 2015; Brožová et al., 2021; Caduff et al., 2022). An automatic deadwood generator which considers the initial stand structure of a forest as well as other relevant drivers of post-disturbance development, may serve as a valuable tool to develop spatially explicit and locally adapted long term scenarios of protection against rockfall after different disturbance and management scenarios. Especially with the advent of area-wide LiDAR data, initial forest conditions are better mapped, and thus more reliable outputs can be expected.

Our modelling results indicate a predominately positive effect of dead wood after natural disturbances without salvage logging. In addition to the medium-term rockfall protection studied here, advanced decay stages of deadwood is beneficial for the biodiversity in particular of saproxylic organisms (Lachat et al., 2013; Sandström et al., 2019), improves biogeochemical cycles and edaphic conditions, provides favourable conditions for tree regeneration and results in greater heterogeneity and functional resilience at different spatial scales (Bače et al., 2012; Shorohova and Kapitsa, 2015; Kulakowski et al., 2017). For the conservation of saproxylic biodiversity Lachat et al. (2013) recommend in their review deadwood densities above $30 \text{ m}^3 \cdot \text{ha}^{-1}$ for mixed mountain forests, but highlights that certain saproxylic species are just found with deadwood densities $\geq 100$ $\text{m}^3 \cdot \text{ha}^{-1}$. While W1 fulfills therefore, the general recommendation, an opportunity was missed in W2 to leave habitat for more demanding saproxylic species and enhance, at the same time, the protective function against natural hazards at least during the first few years after the disturbance. Such an integral look at the effects of deadwood would not be complete without mentioning potential risks associated with leaving deadwood after natural disturbances, including subsequent bark beetle infestations (Stadelmann et al., 2014) and secondary triggering of rockfall after advanced wood decay. Such risks highlight the need for caution in making general recommendations on deadwood management in protection forests and the need for further long-term studies. Nevertheless, our results provide additional evidence and additional arguments for leaving deadwood after natural disturbances in forests with a protective function against rockfall.

While the main programming focus here was set on realistic geometric deadwood patterns, the two step decay function may be further expandable in the future. The different decay rates due to different tree species, elevation (= temperature), exposure, and slope (= soil moisture) collected within the frame of the National Forest Inventories (Hararuk et al., 2020) might be used to interpolate between the two used impact breaking energy states of fresh and roughly 10 year old deadwood (Ammann, 2006)





and could serve to close the gap for simulations a few decades after the disturbance up to the takeover of the post-disturbance
regeneration.

In principle, the ADG could also be used for modelling rockfalls in the context of other natural disturbances, such as bark
beetle infestations and forest fires, with certain adaptations: in the case of a bark beetle infestation, significantly fewer root
plates are expected, and the rot fungi ratio might be reduced due to persistent snugs compared to lying deadwood. However,
the actual direction of fall $\alpha'$ is not easily obtained as it requires continuous monitoring as the process spans over several years
up to decades (Kupferschmid Albisetti et al., 2003; Caduff et al., 2022). Nonetheless, due to the many snags, slope-parallel
deadwood trunk arrangements are to be expected and could be used in a ADG scenario. Similar processes as for bark beetle
infestations are expected for forest fire intensities with low severity and the occurrence of lying and standing, charred trees.
Together with future experimental studies to determine the necessary maximal energy absorption thresholds adjustment, the
ADG could refine first simulation results (Maringer et al., 2016). After forest fires with high severity, the amount of woody
debris is strongly reduced depending on fire severity, and rockfall simulations may after severe fires even assume the *cleared*
condition (Maringer et al., 2016). In addition, the $\mathrm{DSM_{gen}}$ used here only as an intermediate product could also serve as
input for avalanche simulations (Brožová et al., 2021) and the definition of their release areas in forested terrain (Bühler et al.,
2018, 2022). Extending the ADG to other disturbance regimes and including additional time steps of deadwood protection
effects would thus provide a more comprehensive basis for an optimised deadwood management in forests with a protection
function against rockfall.

### 3.6 Sampling ADG Input

Meaningful input parameter sampling is essential in order to use the ADG for further rockfall assessments after large windthrow
events. The ADG requires a DEM, where the accuracy and significance with respect to reality depends on the acquisition date
and processing procedure. Besides, the input parameters of the ADG can be sampled via field excursion and orthophoto
investigation. The additionally necessary tree file can either be meticulously reconstructed on-site, or being generated inside
RAMMS::ROCKFALL with the according forest stand information. Wind throw and root plate ratio are based on visual
inspection and/or recline on previous studies. It arises the question how many deadwood logs $N$ needed to be assessed for
a robust estimate on mean tree fall direction and its standard deviation. The different mean tree fall scenarios per DW-state
in W1 (Fig. 8) differ less than in W2. The smaller influence of the log orientation might have two reasons: a) the overall
$SD$ is larger in W1, which reduces the influence of the main fall direction and b) the overall deadwood density is lower. The
general statistical rule that a population with a lager $SD$ requires a larger sampling size should be questioned, as based on
these results a larger margin of error $e$ for larger (assumed) $SD$ is acceptable. The necessary sampling size $N$ for an assumed
confidence interval of 95% with a z-score $= 1.96$ can be calculated according Eq. 4, where an infinite number of deadwood
logs is assumed.

$$N \le \frac{z^2 + SD^2}{e^2} \tag{4}$$



For example, a sample size of 40 logs and the SD of W1 (48.2) estimates the main wind direction ±15° from the effective wind direction. Ideally, the general deadwood density and DBH-distribution is assessed according pre-storm event data. If the DBH are also assessed from the sampling, the sampling size has to be adapted accordingly.

Even if quasi on-site measurements for input parameters could be available, as herein the wind data from a ridge roughly 4.5 km away, local effects dominate the boundary conditions. The recorded half-hourly averaged wind directions show exclusively an easterly wind direction ($\overline{\alpha}_{IMIS} = 99.9°$, see Fig. 2.b). In contrast, the mapped mean fall direction suggests local northerly ($\overline{\alpha}_{W2} = 7.2°$), respectively north-northeasterly winds ($\overline{\alpha}_{W1} = 26.0°$). Such significant directional differences of 93° in W1, and 74° in W2 should not be neglected, as according Tab. 2 abiding by the IMIS station wind direction would lead to a reduction of $\Delta_{P_R}$ of 14% up to 33%.

## 4 Conclusion

The numerical rockfall simulation taking two natural deadwood log configurations after a storm event into account highlight the influence of lying deadwood on rockfalls with energies smaller than 85 kJ: Run out distances and affected areas are reduced considerably. Fresh deadwood has the highest stopping capacity. The simulations indicate, that even ten year old deadwood pile has a higher protective capacity compared to the pre-storm forest stand. This is a surprising result and confirms that nature-
based protection solutions may further gain in importance in an overall context of economic, logistic and ecological reasons and taking natural hazard risks into account.

In windthrow areas, piled up deadwood logs and uprooted root plates are frequently observed. The pile-up effect in the ADG is reproduced by varying directions of fall for each felled tree sampled from a normal distribution around a mean wind direction and its standard deviation. Aging effects are incorporated via compaction and a reduction of the maximal absorbed
kinetic energy. The created log pattern, its effective height above ground and the added surface roughness due to the deadwood are validated with UAS-field survey data. However, the compaction state is deadwood density dependent and needs to be adapted according to the respective observed or expected prevalent deadwood conditions.

The incorporation of realistic deadwood configurations into the rockfall simulation models helps address urgent questions arising in forestry practice. Automated procedures have been introduced to ease the specification of deadwood location and
pile-up configuration at the slope scale. The approach serves as an effective tool in a post-event analysis. It can be used to determine the necessary amount of lying deadwood that is needed to replace the protection effect of the pre-storm forest and, more importantly, it allows for quantitative benchmarking of different silvicultural measures. The better the fusion of modelling know-how, experimental knowledge of the deadwood stopping capacity, aging effects, and climate models, the more accurate the future protective capacity of a forest can be forecasted.

*Code and data availability.* Upon publication, the ADG, the used orthophoto, and the mapped trees are publicly available under the Envidat repository



*Author contributions.* AdR conceived and developed the deadwood generator, based on the ADG-RAMMS interface and the introduced deadwood-energy-treshold, defined by MC. YB and AS collected the UAS-data. AdR digitized the UAS-data, executed rockfall simulations, evaluated the data set and wrote the draft of manuscript, which was reviewed and edited by AC, PBe, AnR, PBa and YB.

*Competing interests.* The authors declare that they have no conflict of interest.

*Acknowledgements.* The research was partially funded by the National Research Program "Sustainable Economy: resource-friendly, future-oriented, innovative" (NRP 73) by the Swiss National Science Foundation (Grant number: 407340-172415) and is part of the WSL research program Climate Change Impacts on Alpine Mass Movements (CCAMM). Further we thank the Forest and Natural Hazards Division of the Canton Glarus for the collaboration.



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
