# Peer review of "Modelling deadwood for rockfall mitigation assessments in windthrow areas"

_Earth Surface Dynamics, 2022_

## Author Comment (AC1)

**Author's response to the Interactive comment of Referee #1, on "Modelling deadwood for rockfall mitigation assessments in windthrow areas"**

Dear Anonymous Referee #1,

We thank you for the positive conception of our manuscript and for highlighting the relevance of the results for stakeholders in charge of risk management. We present here the answers, which will also be included in the new manuscript version in a suitable form.

The most prominent criticism is the incorrect announce of the deadwood generator use, since forest managers cannot wait for the next windthrow event to get the deadwood logs placed as their mitigation strategy. This is, of course, correct. However, we instead thought about the better quantification of intentionally felled and left logs along the slope as a management strategy. We will clarify this idea within the new manuscript.

The second essential point you have mentioned is the not clearly emphasized conclusion that after windthrow events, a window of opportunity opens for forest managers, as the protective effect of forests against rockfalls is firstly raised. The duration of this increased protection effect is site-specific, as the decay rate is assumed to depend on soil wetness, which depends on precipitation, slope exposition, and temperature, the occurrence of rot fungi and tree species. Further, the amount of deadwood is decisive and the expected rock energies.

Here the follow-up question about the regeneration of the post-windthrow forest has to be discussed, mainly depending on local climate, pre-rejuvenation, and browsing-pressure conditions. But the expected rock energies are also crucial, as dense young forests will do for low energies. For thicker trees, which are necessary to stop mid-to high-energetic rockfalls, growth times are longer, especially in higher elevations.

 Below you will find the response to your remaining criticisms,which we intend to incorporate in the amended manuscript:

*1) Why did you limited the study to one mass class of 400 kg or 0.15 m3? Is it related the block volumes observed on the field?.*

It is a conceivable rock mass for the study site, but it is not based on statistically relevant recording methods. Therefore, we do not highlight this point in the amended manuscript version. However,  we will clarify the following:  Based on experimental studies (Ringenbach et al., 2022), the high protective effect of deadwood against 45 kg rocks is shown. We aimed to extend this rock masses by roughly one order of magnitude.

*2) Line 118 "To directly assess deadwood effectiveness against larger rockfall energies, the fracture impact energy of large-laboratory test data is used": if a simulated rock block impacts a tree with an energy exceeding the fracture impact energy, does it mean that the block will continue to propagate downslope without any disruptions?*

As referee #2 pointed out: the rockfall impacts on deadwood are not described in detail, which led to your question. Your assumption is correct, and blocks exceeding the fracture impact energy propagate without disruptions. We amended the manuscript and added further details of the rockfall impacts on deadwood.

**3) Figure 3: The colors blend. Maybe the orthophoto (background) could be in transparency to improve the reading?**

We used stand-out colors to make the tree trunks easily visible. This color scheme was maybe too much of a treat, and the white background still disturbs it. The adaption of the background transparency might be an option, which we will check for the new manuscript.

---

## Author Comment (AC2)

**Author's response to the Interactive comment of Referee #2, on "Modelling deadwood for rockfall mitigation assessments in windthrow areas"**

Dear Anonymous Referee #2,

We thank you for the overall positive review and the remarks for improvements. We appreciate the label "thoroughly analysed" for the ADG. The more compact presentation of the rockfall module was due to the desire to create a publication that is as concise as possible without repeating too many other papers. Nevertheless, we highly value your outside view and agree that primarily the rock-deadwood interaction was not described in enough detail. The revised manuscript aims to provide more details and amend the objectionable sections. We are convinced that we can fully comply with the referees' main concerns, as an extended description of the rockfall model will be provided and the specifities of the case studies will be addressed and discussed in more details. We strongly believe that a revised manuscript with the already highly appreciated ADG will clearly merit publication in ESURF.

Below you will find the point-by-point answers to the additional questions, which will be appropriately included in the manuscript.

*1) p.3 l. 30: "likely:" → typo.*
We will amend the corresponding sentence.

*2) p. 3 l. 41: the term « absorbed rockfall energy » is not correct in my opinion since the energy of the block is not absorbed, it is converted into kinetic energy of the tree, in particular. This term is thus misleading on the process. I would prefer «block energy reduction».*
Internal discussions about which terms of the cited literature should be used were already held. "Absorbed rockfall energy" is based on Lundstrom (2009) and Kalberer (2007), while Dorren and Berger (2005) are talking of dissipated energy. Both terminologies deviate from the strict definition of absorption initially coined for electromagnetic wave processes. However, in the impact community, energy absorption and, therefore, energy absorbing material, structures etc., are interchangeably used in energy transfer processes between projectile and target (e.g., energy absorbing structures in vehicle crash tests). As "absorbed" energy is widely used for different energy transfer mechanisms, we believe this term remains applicable and is in line with previous publications in the field.

*3) p.4 l.74-80 : The section should be rephrased so that the reader could clearly understand that : the ADG allows to reconstruct deadwood spatial distribution directly after windstorm and also few months after and the rockfall model, depending on the assumptions (modelling of deadwood breakage depending on the level of decay, in particular) allows to assess block propagation in different configurations (i.e. with and without deadwood). In the present form, the sentence could let the reader think that the asssessment of rockfall hazard is mainly depending on the ADG, which is not the case. It should be clearly stated that it depends also on the level of accuracy of the block propagation model.*

We thank for this rephrasing of the ADG introduction as it transports the essence of the paper. We happily amended the section to this effect. We clearly stated that the accuracy of the results depends on the precision of the deadwood parameters as also on the trajectory model parameters.

*4) p. 4 l. 84 : « software » In my opinion, it is not a software, more a python script*
We will use the more detailed "script" instead of software.

*5) p. 4 l. 94 : the variables pw1 and pw2 should be presented here. In adddition, I think that, for more genericity, only a variable pw should be presented.*
We will insert the definition of $p_W$ and the reference to chapter 3.1, where the values of $p_{W1}$ and $p_{W2}$ are specified in more detail.

*6) p. 4 l. 96 : can you justify the values 1.5DBH and 0.65DBH ?*
These values are based on the assumption that the diameter of the root plate is $3 \cdot$ DBH and the stem is $1.3 \cdot$ DBH wide at its base. The values are derived from the mapped root plates and the literature but are deliberately chosen at the lower end to avoid overestimating the protective effect. This adaptation is particularly important since the models assume circular root plates, which may be more oval in reality.

*7) p. 5 l. 108 : can you justify the values 10 % to 20 % : what is the effect of this asssumption?*
We will rearrange the corresponding section for further clarity. We generally suspect that a contact during the fall process in the lowest 10 % of the log will not lead to its final position, as the overturning moment would be too large, and the log will continue to rotate. We disregard the top 20% because it is too flexible, and contact of it on ground (e.g. a cliff) or a preprocessed deadwood log would also not result in a final deadwood position.

*8) p. 5 l. 109 : the H_min values are depending on each study site. What is the effect of this assumption ? Here, it should be stated that there is a potential effect of this value.*
One main result of this study is, that $H_{min}$ values are deadwood density-dependent. We used constant values for fresh deadwood (Hmin =0.645 m) and compacted older deadwood (Hmin =0.258 m) on both sites to obtain this conclusion. These $H_{min}$ values are one-third of the mean observed effective height during field surveys (Bebi et al., 2015). Different effective heights of the deadwood configurations are expected due to the future use of different $H_{min}$ values.

*9) p. 5 l. 119 – 122 : this section let the reader think that the breakage of deadwood is modelled into the rockfall simulations. Is it the case ? If so, the integration of the breakage has to be explained in details.*
To model the breakage of deadwood, a single threshold value based on these reported impact energies is modeled for the former DBH per deadwood log. During rock impacts on deadwood, this threshold is compared to the incoming rock energy: In case the rock has lower kinetic energy than the logs fracture impact energy, the log is taken as a rigid obstacle into account. Otherwise, the deadwood log is neglected for further rock propagation assessment. In terms of a compact model description, we will integrate this clarification into the section 2.3.

**10) section 2,3 : this section has to be substantially improved in order to explain in a more detailled (even if detailled equations are not necessary) way the assumptions associated to the modelling (rigid blocks with a given shape, non-smooth mechanics, rigid trunks and rigid logs,...). In particular, the parameters leading the rebound on the soil have to be explained. The modelling of the impact on the trunk and on the logs is not explained and the associated parameters are not presented. It has to be integrated.**

We rephrased the paragraph, introducing the concept of compactable soils on behalf of the used parameters $M_E$ and $C_d$. We include also the answer to question 9) in this section.

**11) p. 9 l. 180 : the block size is very small. It is a very important asssumption as it strongly influences the efficacity of deadwood. This point has to be clearly presented here but also in the introduction, discussion and conclusion so that the reader can understand the limits of the study.**

We fully agree that the rock size's assumptions must be stated clearly. The original manuscript mentions the rock masses within the abstract and methods section. Additionally, the results and discussion section, as the conclusion, mention the kinetic energies of < 85 kJ, which are linearly dependent on the rock mass but also include the velocity. We think the resulting energy range is even more important, as assessments for different rock masses can also be derived. However, we will incorporate further emphasis on this assumption and its impacts in the amended manuscript.

**12) p. 9 l. 186 : « exemplary visualize » : I don't understand**

Figure 4 visualize**s** exemplary RAMMS::ROCKFALL simulations. We will rewrite this sentence for additional clarity.

**13) p. 9 l. 196 : « 22/79 » means « W1 : 22 , W2 : 79 » ? - to be clarified**

We will clarify the given section and add the corresponding area names W1 and W2.

**14) p. 9 l. 197 :« 2,7-4,9 » : it is not clear that is refers to the slope of the regression – to be clarified**

We will add "The slope of the" to the sentence "linear regression between the DBH and the root plate diameter in W1 and W2 and for the two tree types are in the range between […]".

**15) p. 9 l. 198 : « equitations » - typo**

Thank you for pointing this nasty typo out, which went under the auto-correction radar.

**16) p. 9 l. 199 : At first sight, it is not clear that the range of the regression coefficients are the same (it is a matter of units, I guess) – Am I right ?**

Yes indeed, it was a matter of the units. We will revise the coefficients to achieve the same relationship between the units (DBH in cm due to forestry tradition and root plate diameter in m).

**17) p. 9 l. 204 : Finally, you conclude that it is not necessary to use a regression since the accuracy is low and that a fixed value set at RP_phi=3 DBH is sufficient. Is it correct ? If so, it should be more clearly stated.**

We conclude that in the future, not solely linear regression models should be considered, but, e.g., a normal distribution or other distributions should be checked. Meaning that for each DBH a root plate

diameter according to the chosen distribution is assigned to, as the range for a DBH = 30 cm is quite wide and not well described a unique root plate value.

We left the statement in the original manuscript rather general ("A different, yet to be verified RP distribution"), because we did not investigate the preconditions for the different possible distributions. We clarified the section further and pointed out the primary messages out better.

**18) p. 11 l. 214-215 : this sentence should be moved to a discussion or conclusion section.**

We placed the sentence at the end of a paragraph within the Results and *Discussion* section. Therefore, we are of the humble opinion that such discussions are allowed. Since we have not done further research on an initial machine learning approach, we did not want to put the sentence in conclusion but leave it as an open point in the discussion.

**19) p. 11 l. 219-234 : This section is difficult to follow . Could you improve it ?**

We will rearrange this section in the amended manuscript, aiming for a better reader flow.

**20) p. 15 l. 270-271 : This statement has to be more justified, or explained at least.**

We will add an additionalexplanation sentence: The direction of fall of the deadwood logs has a certain influence on the rockfall retention capacity. This is shown by the altitude of the deposition points between the different scenarios, as depositions at elevations between 1300-1260 meters and 1240-1140 m are solely for the better performing scenarios visible and missing for others.

**21) Figure 8: two last sentences : this point is essential and has to be explained in the main text.**

We also added the train of thought of the zero-point density overshoot in the main text.

**22) p. 15 l. 272-288 : It could have been interesting to compare the distributions of the velocities and energies at given locations of the site (evaluations screens). Indeed, it would give a more straightforward view of the effect of deadwood.**

Indeed the comparison within evaluation screens is a reasonable method to compare different scenarios. The question of how many screens at which location are necessary to depict the complete picture is not straightforward. It requires, in our view, in-depth analysis with preferably a tool implemented directly in the propagation model. As the main scope of this manuscript was the presentation and the proof of concept of the ADG, we have chosen the approach described above, which is more holistic and considers all grid cells but lacks spatial resolution.

**23) p. 15 l. 280-288 : the analysis is based on « energy absorption » which is not relevant , in my opinion. Indeed, as the effect of deadwood depends on many parameters (relative size of blocks and logs, boundray conditions of the logs, incidence of the block), this analysis is too simplified. It is maybe better to suppress it ?**

We agree that there are simplifications in the analysis. However, in our opinion, this is also necessary to recognize the underlying patterns within the large amount of data. We have adjusted the manuscript to draw attention to the simplifications made and point the reader to the overall complexity than deleting the entire section. In addition, this example helps to clarify the limits of deadwood, as requested in question 11. As already mentioned, the rock energy is more meaningful than the rock mass.

**24) p. 16 l. 291 : I would replace « highlight » by « illustrate » since the genericity of the present study is limited.**
We will rephrase the section and will use illustrate instead of highlight.

**25) section 3,5 and conclusion : in these sections, the limitations of the study have to be clearly presented and discussed. In particular, the fact that the block size is small is essential to discuss. In addition, the assumptions of the block progpation model, and their effect on the results, have also to be discussed. Finally, I think that an analysis of the effects of the simulation parameters (that are not presented in the paper) on the mitigation effect of deadwood is missing. This point has to be specified as a perspective , for example**
We enhanced the manuscript at the corresponding sections with either the rock mass or rock energies. A small discussion of the here omitted parameter analysis of $M_E$ and $C_d$ is included in sectin 3.4 referencing to Ringenbach et al., (2022) for the parameter analysis. As the results are quite stable within the given parameter ranges, we do not further investigate this effect in this manuscript. Especially for the predominant forest area, a parameter pair has been used that is covered by the parameter analysis therein.
The last point of the question, which covers an analysis of the effects of the (not presented) simulation parameters, can be answered in two parts: (1.) We will enhance the section about deadwood breakage in the manuscript with insights about the comparison of the incoming kinetic rock energy and the deadwood threshold. (2.) We will emphasize the benefit of a future, detailed calibration experiment with higher energies resulting in the fracture of deadwood.

---

## Author Response (AR1)

**Author's response to the Interactive comment of Referee #1 and #2 on the manuscript "Modelling deadwood for rockfall mitigation assessments in windthrow areas"**

Dear Anonymous Referees,
We thank you for the positive conception of our manuscript and for highlighting the relevance of the results for stakeholders in charge of risk management. Here, we present the answers included in the new manuscript version in a suitable form. In addition to these content adjustments (highlighted in blue in the track change version), we have had the manuscript undergo further detailed English language proofreading by a native speaker. The changes thereof are highlighted in cyan in the track change version.

The most prominent criticism across both referees was the missing extended description of the rockfall model. This is now provided, and the case studies' specificities, such as the deadwood–rock interaction, are addressed and discussed in more detail. We strongly believe this revised manuscript with the already highly appreciated ADG will clearly merit publication in ESURF.

Additionally, referee #1 questioned the incorrect announcement of the deadwood generator use since forest managers cannot wait for the next windthrow event to get the deadwood logs placed as their mitigation strategy. This is, of course, correct. However, we instead thought about better quantifying intentionally felled and left logs along the slope as a management strategy. This idea is within the new manuscript clarified.

The second essential point referee #1 questioned is the not clearly emphasized conclusion that after windthrow events, a window of opportunity opens for forest managers, as the protective effect of forests against rockfalls is firstly raised. The duration of this increased protection effect is site-specific, as the decay rate is assumed to depend on soil wetness, which depends on precipitation, slope exposition, and temperature, the occurrence of rot fungi and tree species. Further, the amount of deadwood is decisive and the expected rock energies. Here the follow-up question about the regeneration of the post-windthrow forest has to be discussed, mainly depending on local climate, pre-rejuvenation, and browsing-pressure conditions. But the expected rock energies are also crucial, as dense young forests will do for low energies. For thicker trees, which are necessary to stop mid-to high-energetic rockfalls, growth times are longer, especially in higher elevations.

Below you will find the response to the remaining criticisms, starting with referee #1 and followed by referee #2.

**Referee #1**

*1) Why did you limited the study to one mass class of 400 kg or 0.15 m3? Is it related the block volumes observed on the field?.*

It is a conceivable rock mass for the study site, but it is not based on statistically relevant recording methods. Therefore, we do not highlight this point in the amended manuscript version. However, based on experimental studies (Ringenbach et al., 2022), the high protective effect of deadwood against 45 kg rocks is shown. We aimed to extend this rock masses by roughly one order of magnitude.

*2) Line 118 "To directly assess deadwood effectiveness against larger rockfall energies, the fracture impact energy of large-laboratory test data is used": if a simulated rock block impacts a tree with an energy exceeding the fracture impact*

*energy, does it mean that the block will continue to propagate downslope without any disruptions?*

As referee #2 pointed out: the rockfall impacts on deadwood are not described in detail, which led to your question. Your assumption is correct, and blocks exceeding the energy threshold propagate without disruptions. We amended the manuscript and added further details of the rockfall impacts on deadwood (l. 128 ff. and l. 179-208).

*3) Figure 3: The colors blend. Maybe the orthophoto (background) could be in transparency to improve the reading?*

We used standout colors to make the tree trunks easily visible. This color scheme was maybe too much of a treat, and the white background still disturbs it. The adaption of background transparency was checked as an option but did not yield better results. Taking into account the possibility that the figure can be viewed in full size and so the minor problems are completely fixed, we have left the figure as it was.

**Referee #2**

*1) p.3 l. 30: "likely:" → typo.*
We amended the corresponding sentence (p.2, l. 32).

*2) p. 3 l. 41: the term « absorbed rockfall energy » is not correct in my opinion since the energy of the block is not absorbed, it is converted into kinetic energy of the tree, in particular. This term is thus misleading on the process. I would prefer «block energy reduction».*
Internal discussions about which terms of the cited literature should be used were already held. "Absorbed rockfall energy" is based on Lundstrom (2009) and Kalberer (2007), while Dorren and Berger (2005) are talking of dissipated energy. Both terminologies deviate from the strict definition of absorption initially coined for electromagnetic wave processes. However, in the impact community, energy absorption and, therefore, energy absorbing material, structures etc., are interchangeably used in energy transfer processes between projectile and target (e.g., energy absorbing structures in vehicle crash tests). As "absorbed" energy is widely used for different energy transfer mechanisms, we believe this term remains applicable and is in line with previous publications in the field.

*3) p.4 l.74-80 : The section should be rephrased so that the reader could clearly understand that : the ADG allows to reconstruct deadwood spatial distribution directly after windstorm and also few months after and the rockfall model, depending on the assumptions (modelling of deadwood breakage depending on the level of decay, in particular) allows to assess block propagation in different configurations (i.e. with and without deadwood). In the present form, the sentence could let the reader think that the asssessment of rockfall hazard is mainly depending on the ADG, which is not the case. It should be clearly stated that it depends also on the level of accuracy of the block propagation model.*

We thank for this rephrasing of the ADG introduction as it transports the essence of the paper. We happily amended the section to this effect. We clearly stated that the accuracy of the results

depends on the precision of the deadwood parameters as also on the trajectory model parameters. (p.4, l. 77 – 83)

**4) p. 4 l. 84 : « software » In my opinion, it is not a software, more a python script**
We will use the more detailed "script" instead of software. P.4, l. 91

**5) p. 4 l. 94 : the variables pw1 and pw2 should be presented here. In adddition, I think that, for more genericity, only a variable pw should be presented.**
We inserted the definition of $p_W$ and the reference to chapter 3.1, where the numerical values of $p_{W1}$ and $p_{W2}$ are specified in more detail. (p. 4, l. 101)

**6) p. 4 l. 96 : can you justify the values 1.5DBH and 0.65DBH ?**
These values are based on the assumption that the diameter of the root plate is $3 \cdot DBH$ and the stem is $1.3 \cdot DBH$ wide at its base. The values are derived from the mapped root plates and the literature but are deliberately chosen at the lower end to avoid overestimating the protective effect. This adaptation is particularly important since the models assume circular root plates, which may be more oval in reality. (p. 4, l. 103 – p5. l. 106)

**7) p. 5 l. 108 : can you justify the values 10 % to 20 % : what is the effect of this asssumption?**
We rewrote the corresponding section for further clarity. We generally suspect that a tree contact within the lowest 10 % of the falling/rotating stem will not lead to its final position, as the overturning moment would be too large, and the log would continue to turn. We disregard the top 20% because it is too flexible, and contact with it on the ground or preprocessed deadwood logs would also not result in a final deadwood position. (p.5 l. 116 – 119)

**8) p. 5 l. 109 : the H_min values are depending on each study site. What is the effect of this assumption ? Here, it should be stated that there is a potential effect of this value.**
One main result of this study is, that $H_{min}$ values are deadwood density-dependent. To obtain this conclusion, we used constant values for fresh deadwood ($H_{min}$ =0.645 m) and compacted older deadwood ($H_{min}$ =0.258 m). These $H_{min}$ values are one-third of the mean observed effective height during field surveys (Bebi et al., 2015, p. 5, l. 119). Additionally, the effect of the $H_{min}$ is discussed and an adaption strategy for lower deadwood densities is stated (p. 13. l. 275).

**9) p. 5 l. 119 – 122 : this section let the reader think that the breakage of deadwood is modelled into the rockfall simulations. Is it the case ? If so, the integration of the breakage has to be explained in details.**
The breakage and its energy absorption itself are not modeled. However, the deadwood is considered as an obstacle with hard contact laws up to a single energy threshold value per deadwood log. This threshold is calculated based on the DBH of the hitherto standing tree and the fracture impact energies of Ammann (2006). During rock impacts on deadwood, this threshold is compared to the incoming rock energy: In case the rock has lower kinetic energy than the logs fracture impact energy, the log is taken as a rigid obstacle into account. Otherwise, the deadwood log is wholly neglected for the rock propagation assessment. (l. 128 ff. and l. 179-208)

**10) section 2,3 : this section has to be substantially improved in order to explain in a more detailled (even if detailled equations are not necessary) way the assumptions associated to the modelling (rigid blocks with a given shape, non-smooth mechanics, rigid trunks and rigid logs,…). In particular, the parameters leading the rebound on the soil have to be explained. The modelling of the impact on the trunk and on the logs is not explained and the associated parameters are not presented. It has to be integrated.**

We rephrased the paragraph, introducing the concept of compactable soils on behalf of the used parameters $M_E$ and $C_d$. As the answer to question 9) includes the new approach on how the breakage of deadwood is considered, it is not repeated (l. 179-208).

**11) p. 9 l. 180 : the block size is very small. It is a very important asssumption as it strongly influences the efficacity of deadwood. This point has to be clearly presented here but also in the introduction, discussion and conclusion so that the reader can understand the limits of the study.**

We fully agree that the rock size's assumptions must be stated clearly. The original manuscript mentions the rock masses within the abstract and methods section. Additionally, the results and discussion section, as the conclusion, mention the kinetic energies of < 85 kJ, which are linearly dependent on the rock mass but also include the velocity. We think the resulting energy range is even more important, as assessments for different rock masses can also be derived. Nevertheless, we enriched the conclusion with the used rock masses (p. 21, l. 417)

**12) p. 9 l. 186 : « exemplary visualize »: I don't understand**

We changed the sentence to "are visualized in Figure 4 as an example".

**13) p. 9 l. 196 : « 22/79 »  means « W1 : 22 , W2 : 79 » ? - to be clarified**

We will clarified the given section and add the corresponding area names W1 and W2.    (p. 10, l. 234)

**14) p. 9 l. 197 :« 2,7-4,9 » : it is not clear that is refers to the slope of the regression – to be clarified**

We will add "The slopes of the" to the sentence "linear regressions between the DBH and the root plate diameter in W1 and W2 and for the two tree types are in the range between […]. (p. 10, l. 235)

**15) p. 9 l. 198 : « equitations » - typo**

Thank you for pointing this nasty typo out, which went under the auto-correction radar.

**16) p. 9 l. 199 : At first sight, it is not clear that the range of the regression coefficients are the same (it is a matter of units, I guess) – Am I right ?**

Yes indeed, it was a matter of the units. We revised the coefficients to achieve the same relationship between the units (DBH in cm due to forestry tradition and root plate diameter in m).

**17) p. 9 l. 204 : Finally, you conclude that it is not necessary to use a regression since the accuracy is low and that a fixed value set at RP_phi=3 DBH is sufficient. Is it correct ? If so, it should be more clearly stated.**

We conclude that in the future, not solely linear regression models should be considered, but, e.g., a normal distribution or other distribution should be checked. We left the statement in the original

manuscript rather general ("A different, yet to be verified RP distribution"), because we did not investigate the preconditions for the different possible distributions. We clarified the section further and pointed out the primary messages out better. (p. 10, l 244- 246).

**18) p. 11 l. 214-215 : this sentence should be moved to a discussion or conclusion section.**
We placed the sentence at the end of a paragraph within the *Results and Discussion* section. Therefore we are of the humble opinion that such discussions are allowed. Since we have not done further research on an initial machine learning approach, we did not want to put the sentence in conclusion but leave it as an open point in the discussion.

**19) p. 11 l. 219-234 : This section is difficult to follow . Could you improve it ?**
We enhanced the given section and focused on better readability and flow.

**20) p. 15 l. 270-271 : This statement has to be more justified, or explained at least.**
We added an additional explanation sentence: The direction of fall of the
deadwood logs has thus a certain influence on the rockfall retention capacity, as depositions at these elevations are solely for the better performing scenarios visible. (p.15, l 312).

**21) Figure 8: two last sentences : this point is essential and has to be explained in the main text.**
We also added the train of thought of the zero-point density overshoot in the main text. (p.15, l 316-318).

**22) p. 15 l. 272-288 : It could have been interesting to compare the distributions of the velocities and energies at given locations of the site (evaluations screens). Indeed, it would give a more straightforward view of the effect of deadwood.**
Indeed the comparison within evaluation screens is a reasonable method to compare different scenarios. The question of how many screens at which location are necessary to depict the complete picture is not straightforward. It requires, in our view, in-depth analysis with preferably a tool implemented directly in the propagation model. As the main scope of this manuscript was the presentation and the proof of concept of the ADG, we have chosen the approach described above, which is more holistic and considers all grid cells but lacks spatial resolution.

**23) p. 15 l. 280-288 : the analysis is based on « energy absorption » which is not relevant , in my opinion. Indeed, as the effect of deadwood depends on many parameters (relative size of blocks and logs, boundray conditions of the logs, incidence of the block), this analysis is too simplified. It is maybe better to suppress it ?**
We agree that there are simplifications in the analysis. However, in our opinion, this is also necessary to recognize the underlying patterns within the large amount of data. We have adjusted the manuscript to draw attention to the simplifications made and point the reader to the overall complexity than deleting the entire section. In addition, this example helps to clarify the limits of deadwood, as requested in question 11. As already mentioned, the rock energy is more meaningful than the rock mass.

**24) p. 16 l. 291 : I would replace « highlight » by « illustrate » since the genericity of the present study is limited.**
We rephrased the section and used illustrate instead of highlight (p., l. 340).

**25) section 3,5 and conclusion : in these sections, the limitations of the study have to be clearly presented and discussed. In particular, the fact that the block size is small is essential to discuss. In addition, the assumptions of the block progpation model, and their effect on the results, have also to be discussed. Finally, I think that an analysis of the effects of the simulation parameters (that are not presented in the paper) on the mitigation effect of deadwood is missing. This point has to be specified as a perspective , for example**
We enhanced the manuscript in the corresponding sections with either the rock mass or rock energies. A small discussion of the omitted parameter analysis of $M_E$ and $C_d$ is included in section 3.4 (p.18, l. 335), referencing Ringenbach et al. (2022) for the parameter analysis. As the results are relatively stable within the given parameter ranges, we do not further investigate this effect in this manuscript. Especially for the predominant forest area, a parameter pair has been used that is covered by the parameter analysis therein.
The last point of the question, which covers an analysis of the effects of the (not presented) simulation parameters, can be answered in two parts: (1.) We enhanced the section about deadwood breakage in the manuscript with insights about the comparison of the incoming kinetic rock energy and the deadwood threshold. (l. 128 ff. and l. 179-208) (2.) We emphasized the benefit of a future, detailed calibration experiment with higher energies resulting in the fracture of deadwood (p. 21, l. 437-439).